# Immune Regulatory Functions of Macrophages and Microglia in Central Nervous System Diseases

**DOI:** 10.3390/ijms24065925

**Published:** 2023-03-21

**Authors:** Michael Poppell, Grace Hammel, Yi Ren

**Affiliations:** Department of Biomedical Sciences, Florida State University College of Medicine, Tallahassee, FL 32306, USA

**Keywords:** multiple sclerosis, Alzheimer’s disease, Parkinson’s disease, spinal cord injury, traumatic brain injury, immune response, microglia, macrophage

## Abstract

Macrophages can be characterized as a very multifunctional cell type with a spectrum of phenotypes and functions being observed spatially and temporally in various disease states. Ample studies have now demonstrated a possible causal link between macrophage activation and the development of autoimmune disorders. How these cells may be contributing to the adaptive immune response and potentially perpetuating the progression of neurodegenerative diseases and neural injuries is not fully understood. Within this review, we hope to illustrate the role that macrophages and microglia play as initiators of adaptive immune response in various CNS diseases by offering evidence of: (1) the types of immune responses and the processes of antigen presentation in each disease, (2) receptors involved in macrophage/microglial phagocytosis of disease-related cell debris or molecules, and, finally, (3) the implications of macrophages/microglia on the pathogenesis of the diseases.

## 1. Introduction

Within the central nervous system (CNS), there are several cell types that have the ability to act as antigen-presenting cells (APCs) by first phagocytosing materials and then presenting them to lymphocytes. Some of these non-professional APCs include endothelial cells [1,2,3,4], pericytes [5,6,7], and astrocytes [8,9]. However, in this manuscript, we will focus on microglia and infiltrated macrophages, the professional phagocytes that can be found within the CNS under physiological conditions. Within the different disease models discussed in this review, both inflammation and a subsequent immune response are highly characterized, thus offering an interesting insight into the many alternative functions macrophages and microglia can play during these CNS pathologies. One of these roles may include presenting self-antigens to lymphocytes and initiating adaptive immune response (Figure 1). Autoantibodies, or antibodies that react with self-antigens, are typically associated with autoimmune disorders. However, they have been observed in various neurodegenerative disorders and CNS trauma and have rapidly become an area of interest as potential diagnostic markers and therapeutic targets [10,11]. Here, we will discuss the roles that macrophages and microglia play in various CNS diseases by presenting information regarding their phagocytic and antigen-presenting capabilities in different disease types and what implications this may have on the pathogenesis of each disease.

## 2. Neurotrauma

Trauma to any part of the CNS can cause a plethora of debilitating symptoms as well as a substantial immune response. Worldwide, 2.5 million individuals are estimated to sustain some form of traumatic brain injury (TBI) each year, with approximately 250,000–500,000 new cases of spinal cord injury (SCI) occurring each year, globally [12,13,14]. SCIs are a debilitating condition that results in a myriad of symptoms, ranging from disruption of sensory functions to tetraplegia, and can be caused by various external insults such as motor vehicle accidents and falls [15]. Traumatic SCI initially occurs with a primary injury phase initiated by a physical insult that damages the spinal cord [16,17]. After the initial primary injurious event of SCI, a secondary phase of injury begins 2–48 h afterward in which the CNS inflammatory response to the injury causes further damage through mechanisms such as free radical production, lipid peroxidation, inflammation, and necrosis [18]. TBIs are characterized as injuries caused to the brain by external forces such as car accidents or sports related injuries. Symptoms associated with TBI can include post-traumatic seizures and agitation, balance disorders, major depression, anxiety, and aggression [19]. TBI causes CNS damage through mechanisms such as neuroinflammation, oxidative stress, and mitochondrial dysfunction [12,20]. Microglia and peripheral macrophages are integral cellular mediators of inflammation in both SCI and TBI [21,22].

### 2.1. Immune Response and Antigen Presentation in Neurotrauma

Although there are numerous self-reactive autoantigens that are present is the various disease states covered in this review, in Table 1, we present several of the most commonly studied autoantibody antigens. Currently, several different autoantibodies have been shown to exist at significantly increased levels in neurotrauma patients. For example, myelin basic protein (MBP) autoantibodies were found in significantly elevated levels in both SCI and TBI, with the anti-MBP autoantibody levels being used as a marker for severity and outcome in TBI patients [23,24]. Though there appear to be elevated levels of autoantibodies to GM1 (a ganglioside that is primarily found in neurons) in the sera of individuals with SCI, available data are conflicting [25]. While one study found a significant increase in immunoglobulin M (IgM) antibody to GM1 but not IgG antibody to GM1 [26], another found significantly elevated IgG levels but not IgM to GM1 [27]. In SCI patient sera, anti-MAG (myelin-associated glycoprotein) antibodies were not found to be elevated as compared to healthy controls [26,27]. Elevated levels of autoantibodies to glial fibrillary acidic protein (GFAP), collapsin response mediator protein-2 (CRMP2), and MBP have also been reported in the plasma or sera of SCI patients [28,29,30,31]. As reviewed by Needham et al., many autoantibodies have been studied in TBI patients, with targets that include glutamate receptors, phospholipids, and acetylcholine receptors [32]. The autoantibody targets that are present in the greatest number of TBI patients are autoantibodies against β-tubulin class III (βTcIII), GFAP, and S100B [32]. S100B, an astrocytic protein, was the highest in serum samples of football players, which correlated with the number of repeated head hits and their intensity during games [33]. Serum IgG against βTcIII, which is found in neuronal cytoskeletons, was significantly elevated over that of healthy controls from 21–23 days post-injury until up to day 30; however, it was noted that there was high variability amongst the subjects [34]. Further studies of sera of TBI patients indicated that patients with chronic TBI had significantly higher levels of autoantibody against GFAP, suggesting that autoantibody production is increased in more severe cases [35,36]. Given the limited information currently available, further research is necessary to determine the potential roles of these autoantibodies’ in neurotrauma pathology and their potential as therapeutic targets.

The molecules of major histocompatibility complex (MHC) class I and II are upregulated in macrophages and microglia near the SCI lesion area after injury [37,38,39,40]. It has been reported that there is an influx of T lymphocytes into the lesion area after SCI, with one study showing elevated levels of both CD4^+^ T cell and B cell responses in the sub-acute phase post-SCI [37,39,73]. T cells from blood samples of SCI patients were found to be reactive to MBP, with data showing that the reactive T cells are CD4^+^ [23,74,75]. Increased expression of MHC I and II has been found after TBI; however, whether the increase is significant or not appears to depend on multiple factors [43,44,45]. For example, a significant increase in MHC I and MHC II gene expression was observed in the ventral posteromedial nucleus of the thalamus (VPM) but is only elevated in the primary somatosensory barrel field (S1BF) [43]. MHC II expression was elevated on microglial/macrophage in the cortex of young (3-month-old) and aged (24-month-old) mice, but it was only a significant elevation in aged mice after TBI [44]. Infiltration of T cells at the TBI lesion area occurs around 24 h post-injury and dissipates by 7 days post-injury [76,77]. Available evidence also suggests that CD8^+^ T cells are the main immune cells to infiltrate the lesion area [76,78,79]. In one study, utilizing single-cell RNA and T cell receptor sequencing, a population of CD8^+^ T lymphocytes were found in the lesion brain tissues of patients with radiation-induced brain injury [80]. They also determined that microglia-derived chemokines mediated the infiltration of these CD8^+^ T cells and that this chemotactic interaction played a crucial role in disease onset and progression of radiation-induced brain injury [80]. Gaining further understanding of the multifaceted and robust immune response after neurotrauma is critical in fully recognizing the potentially pathogenic and disease progressing roles many of these immune cells play. As described above, macrophages and microglia are responsible for presenting antigens to T lymphocytes after injury, and this may also contribute to the production of the various autoantibodies in patients after neurotrauma. To further investigate the function of macrophages and microglia in neurotrauma pathology, we will discuss the receptors by which these cells uptake cell debris after injury, leading to their activation and subsequent antigen presentation.

### 2.2. Receptors Involved in Macrophage/Microglial Phagocytosis of Myelin Debris in Neurotrauma

Various studies have investigated macrophage and microglia phagocytosis of cellular debris after SCI. Microglia are the first phagocytes to surround axons after SCI until 3 days post-injury, at which point macrophages that have infiltrated from the periphery become the dominant phagocytes [21]. Macrophages, once having arrived at the lesion center after SCI, express a number of surface receptors to mediate the uptake of various forms of cellular debris, including debris formed from myelin sheaths, which we will focus on in this section. The process of macrophage phagocytosis in general involves the binding of particles to receptors, which then triggers the internalization of the particles into an early phagosome, which then matures into a late phagosome [81]. The phagosome then merges with a lysosome and forms the phagolysosome, during which time the materials within the phagolysosome are digested [81]. Macrophage phagocytosis of myelin debris is facilitated by complement receptor 3 (CR3; also known as Mac-1, integrin αMβ2 or CD11b/CD18), scavenger receptor AI/AII (SR-AI/II), TREM2, MARCO, and Fc-receptors [48,49,50,82,83,84,85,86]. C1q, which complexes with CR3, may play a role in mediating microglia clearance of axons after SCI [21,41]. Microglia in rats with SCI also show upregulation of Mac-2 (galectin-3), which may also be involved in phagocytosis of myelin debris [42]. Through the use of a C3 inhibitor that attenuates C3d opsonization in vivo, microglia and potentially macrophage-mediated phagocytosis of hippocampal synapses are inhibited in mice subjected to TBI [46]. The mRNA expression in brain tissue from TBI mice show upregulation of *Cd45*, *MhcII*, and *Cd68*, which are the markers for microglia and macrophage activation and phagocytosis [47].

### 2.3. Implications in Pathogenesis of Neurotrauma

In both SCI and TBI, clearance of cellular debris, including myelin debris, is critical for generating a pro-regenerative environment and is a prerequisite for injury repair; however, the uptake of myelin debris by macrophages can lead to various deleterious effects on the injury microenvironment. Macrophages can take on a spectrum of phenotypes from pro-inflammatory M1-like phenotypes to anti-inflammatory M2-like phenotypes. M1 polarized microglia and macrophages further contribute to injury progression through secretion of TNF-α and Il-1β and production of ROS and NOS [22,87]. Both TNF-α and Il-1β are important cytokines in activating the innate immune response, mediating the recruitment as well as activation of immune cells such as macrophages, and have potent proinflammatory capabilities; however, their dysregulation can lead to a plethora of pathological conditions including the chronic inflammation seen after neurotrauma [88]. TNF-α signaling is initiated by TNF-α binding to its receptors TNF receptor 1 (TNFR1) and TNF receptor 2 (TNFR2), with TNFR1 mainly stimulating proliferation as well as cell survival via nuclear factor-kappa B (NF-κB) signaling and mitogen-activated protein kinase (MAPK) signaling, as well cell apoptosis via activation of caspase 3 [89]. TNFR2, which is found in limited cell types such as endothelial cells and immune cells, is primarily associated with inflammation and cell survival [90]. TNF-α binding to TNFR2 leads to the recruitment of TNF receptor-associated factor (TRAF) 2, TRAF3, TRAF1, and cellular inhibitor of apoptosis protein (cIAP) cIAP1/2 proteins within the cell to form a complex that activates MAPK kinase kinase-3 (MEKK-3), leading to NF-κB generation [90]. TNF-α binding to TNFR2, similarly to TNFR1, triggers apoptosis signaling kinase-1 (ASK-1), which eventually results in p38 MAPK activation, and subsequent activation of activator protein 1 (AP-1) [90,91]. Additionally, TNFR2 activation can lead to its interaction with endothelial/epithelial tyrosine kinase (Etk), which subsequently activates vascular endothelial growth factor receptor 2 (VEGFR2) [90]. VEGFR2 activation then leads to the activation of the (PI3K)/Akt pathway, which regulates processes such as cell adhesion, cell survival, and proliferation [92]. Il-1β binds to IL-1 receptor accessory protein (IL-1RAcP), which then recruits myeloid differentiation primary response protein 88 (MyD88), IL-1R associated kinase 4 (IRAK4), and TRAF6, which causes the activation of NF-κB and MAPK, resulting in the transcription of several inflammatory genes [93]. M2 polarized microglia and macrophages have been shown to promote CNS repair and limit secondary inflammatory mediated injury by producing anti-inflammatory cytokines [22,87]. In both cases of SCI and TBI, the population of M2-like microglia and macrophages decreases within a few days post-injury, while the M1-like activated cells persist for extended periods of time [22,84,94]. With macrophage phenotypes existing on a spectrum of disease-promoting to tissue regenerative, studies have shown they often serve a dichotomous role in disease pathogenesis. Specifically, in SCI, bone marrow-derived macrophages (BMDMϕ) infiltrate to the epicenter of the injured spinal cord where they phagocytose myelin debris to become myelin-laden macrophages (Mye-Mϕ), which are closely associated with the pro-inflammatory signaling cascades predominant in SCI [94,95,96,97,98,99,100,101]. Myelin debris phagocytosis also leads to decreased phagocytic capacity of macrophages, as well as dampening of their response to future stimuli [96,102,103]. Furthermore, our group demonstrated that the expression of neuron-glial antigen 2 (NG2) was increased in Mye-Mϕ, and NG2^+^ Mye-Mϕ displayed enhanced proliferation and decreased phagocytic capacity [103]. The decreased phagocytic capacity in Mye-Mϕ could lead to decreased clearance of necrotic/apoptotic cells after injury, further promoting neurotoxicity, tissue damage, and secondary injury progression [94,96,98,99,100,101,104]. For a complete review of the beneficial and detrimental effects of myelin debris phagocytosis by macrophages in SCI as well as macrophage’s contribution to SCI pathology see [82,98]. Given that M1 polarized microglia and macrophages appear to dominate the cellular milieu of TBI and SCI-affected lesion areas, modulating the M2 response could be a potential target for therapeutics and may be key to promoting resolution to inflammation and tissue healing after injury. For further review on myelin debris phagocytosis by cells of the CNS and consequences of myelin debris uptake see [83].

Administration of IL-13, an anti-inflammatory cytokine, in mice subjected to TBI enhances microglia phagocytic activity and association of microglia with damaged neurons [105,106]. In rats that were subjected to brain injury, neurons near the cortical lesion area strongly colocalized with IgG, peaking in intensity 24 h after lesion induction [107]. Further examination of these neurons showed that many of them were undergoing cell death, implying that the IgG binding can facilitate phagocytosis and clearance via Fc receptor expressing cells [107]. Further research is warranted to study the behavior of microglia and macrophages and their phagocytic interactions under TBI conditions. A recent study performed on IgM knock-out (KO) mice showed that after cervical SCI, IgM KO mice exhibited worse outcomes compared to their wild-type (WT) controls [108]. The IgM KO mice also had elevated T cell infiltration 2 weeks post-injury and significant loss of neural tissue at 10 weeks post-injury [108]. The recovery of locomotor function after SCI was improved in recombination activating gene 2 (*RAG2*) depletion mice (*Rag2^−/−^*), in which they lacked mature T cells and B cells, as compared to controls, with a reduction in microglia/macrophages, implying that adaptive immunity plays a deleterious role post-injury [109]. Interestingly, there is no difference in the extent of injury severity, inflammation, tissue damage, neuronal cell death, and neurological impairment between recombination activating gene 1 (*RAG1*) depleted mice (*Rag1^−/−^*), which also lack mature B cells and T cells, and WT mice after TBI, suggesting that adaptive immunity mediated by T and B cells may not play a critical role in initiating and sustaining the neuroinflammatory response seen after TBI [110]. Conversely, after SCI in a B-cell lacking mouse line, the locomotor recovery was improved and lesion pathology was reduced as compared to controls [111]. In the same study, B cells, IgG, and IgM were found in the cerebrospinal fluid and injured spinal cord of WT mice but were not present in the B cell-lacking mice, leading to a large accumulation of antibody and activation of complements in the injury area of the WT mice. Overall, this study demonstrates that humoral immunity may play a significant role in SCI pathology. Taken together, further studies must be performed to completely understand the role of antibody-mediated immune response in SCI and TBI injury progression and how these roles may differ between the two injury types.

In summary, numerous autoantibodies have been identified in both TBI and SCI, some unique to each injury and others shared between the two. A range of receptors has been shown to mediate the phagocytosis and clearance of cell debris such as myelin debris by macrophages/microglia. Following injury, both M1-like disease-promoting and M2-like neuroprotective microglia and peripheral macrophages are present, but during the chronic stage, M1-like macrophages/microglia are the predominant phenotype in the injury area. Striking a balance between the neuroprotective and pro-inflammatory responses in response to neurotrauma appears to be a promising path of research that could uncover novel therapeutic options for patients.

## 3. Multiple Sclerosis

Multiple sclerosis (MS) is a chronic autoimmune disease resulting in the degradation of the critical myelin sheath of neuronal axons in the CNS [112]. In a recent comprehensive global epidemiologic study regarding MS, approximately 2.8 million people are estimated to live with MS worldwide [113,114]. The pathology of MS is characterized by lesions throughout the white matter of the CNS that exhibit demyelination, inflammation, and glial reaction [115]. The cellular profile of these lesions is composed of both activated macrophages, microglia, as well as T and B cells that have migrated from the periphery after disruption of the blood–brain barrier [116]. Symptoms of MS include, but are not limited to, acute optic neuritis, palsies, ataxia, sensory deficits, and cognitive impairment [117].

### 3.1. Immune Responses and Antigen Presentation in MS

Given that MS is a demyelinating disease, autoantibodies specific to myelin and myelin-proteins have long been targets in the search for reliable biomarkers and are the major autoantigen candidates in MS [51]. Multiple studies have demonstrated that autoantibodies for MBP, a multifunctional protein that composes a significant portion of CNS myelin protein, can reliably be detected in the CSF of patients diagnosed with MS, while only 2% of samples amongst non-MS controls had MBP autoantibodies [118,119,120,121]. While MBP autoantibodies are reliably detectable at elevated levels in the CSF of MS patients, studies of sera have shown conflicting results. One group detected elevated autoantibodies to MBP in 25% of MS patients and 10% of controls, while a different group found elevated autoantibodies in 77% of MS patients and only 5% of controls [51,122,123]. Further research into the presence, or lack thereof, of MBP autoantibodies in patient sera is required based on these findings.

Myelin oligodendrocyte glycoprotein (MOG) has long been another target of MS auto-antibody research since it resides in the outermost surface of the myelin sheath surrounding neurons [124]. Overall findings from studies concerning the presence of MOG autoantibodies in MS patients have been conflicting. MOG autoantibodies were detected in 50% of brain tissue samples of postmortem MS cases, but none in the control tissue that did not have other neuroinflammatory conditions [124,125]. When testing MS patient sera, one study found only approximately 5% of MS patients tested positive for anti-MOG IgG [126]. Using blood sera from patients with primary progressive or secondary progressive MS, MOG-IgG autoantibodies were not detectable amongst both populations [127]. Contradictory findings were shown that detected significant differences in IgM and IgG to MOG in the sera of patients with MS as compared to controls [128]. In light of these conflicting results and sporadic detection of MOG amongst different MS groups, research into more refined detection methods of MOG may provide better answers.

Since the major hallmark of MS is demyelination within the CNS, the main focus of autoantibody research has been on myelin antigens such as MBP and MOG. However, there has been some success in finding autoantibodies to the lipid components of myelin, but the current biomarkers and their diagnostic assays do not appear to match those of MBP or MOG so far in either sensitivity or specificity [51,129,130]. As reviewed by Fraussen et al., there is active research into autoantibodies in MS to components of neurons and axons, oligodendrocytes, astrocytes, immune cells, as well as antibodies to viral antigens and antibodies to ubiquitous antigens such as heat shock proteins [51].

One survey of MS lesions found CD8^+^ T cells and CD20^+^ B cells to predominate across all disease and lesion stages, while CD4^+^ T cells were sparse [131]. As reviewed by Liu et al., Th1 and Th17 lymphocytes appear to be the primary mediators of damage to myelin sheaths in MS, and B cells that produce autoantibodies to myelin protein have been observed in MS patients [132]. The cacophony of chemokines and cytokines that are produced during MS is most likely responsible for the recruitment of Th1 and Th17 CD4^+^ T cells, CD8^+^ T cells, and B cells that further contribute to CNS damage in MS and experimental autoimmune encephalomyelitis (EAE), an animal model for MS [131,133,134,135].

The increase in MHC I and MHC II expression was consistently observed around MS lesions in brain samples from MS patients. While other cells such as astrocytes and endothelial cells in the brain express these molecules, microglia and macrophages are the strongest MHC expressing cells [136,137,138]. Microglia and macrophages in brain tissue of MS patients also express the costimulatory molecule family B7, which includes B7-1 (CD80) and B7-2 (CD86) [139,140,141]. B7 serves as a costimulatory molecule for CD28/CTLA-4, in concert with binding to MHC II to activate CD4^+^ T cells [139]. B7 plays a critical role in the initiation and expansion of MOG-reactive T cells in EAE, and macrophage signaling through B7 molecules may be critical to the activation, as well as regulation, of encephalitogenic T cells [142,143]. These findings interestingly implicate a significant contribution of CD4^+^ T cells in the effector phase of EAE and possibly MS.

CD40, an MHC II costimulatory molecule, and its associated ligand CD40 ligand (CD40L; CD154), which is found on activated T cells, have been theorized to participate in immune-mediated pathogenesis of MS [144]. Peripheral blood samples, but not CSF, from patients with secondary progressive MS have constitutive CD40L expression on CD4^+^ and CD8^+^ T cells, indicating some form of systemic immune response in MS [145]. While significantly higher than control groups, the median value of CD4^+^ T cells expressing CD40L was 3.4%, and only 1.7% of CD8^+^ T cells express CD40L [145]. Colocalization of CD40 expressing microglia and macrophages with CD40L expressing CD4^+^ T cells has been observed in post-mortem sections of brain tissue from MS patients, implying that MHC II and CD40 mediated antigen presentation between microglia/macrophages and CD4^+^ T cells is taking place [146]. When compared to WT controls, EAE mice in which CD40 expression is specifically depleted in macrophages, exhibited significantly lower disease severity, a reduction in neuroinflammation, and decreased myelin debris phagocytosis [147]. Upon induction of EAE in CD40^−/−^ bone marrow chimeric mice that have CD40-deficient microglia but maintained WT peripheral macrophages, the mice exhibited very mild or no disease symptoms compared to WT controls [148,149]. Macrophage recruitment in the CNS at the peak of EAE progression was also lower in the CD40^−/−^ chimeric mice as compared to WT mice [148,149]. A lack of neuroinflammation has also been reported in these CD40^−/−^ chimeric mice after EAE induction, as well as in mice treated with anti-CD40L antibody [144,150]. Taken together, these studies suggest that both microglia and macrophages contribute to MS disease progression in a complementary fashion and that no one aspect is solely responsible for demyelination or neuroinflammation.

Another costimulatory molecule, CD137, is potentially involved in mediating the immune response in MS, but information is lacking regarding the demonstration of cellular expression of CD137 in situ [144]. CD137 expression is significantly increased in CD8^+^ T cells in white matter lesions of brain tissue from MS patients, suggesting that CD137 expression on microglia may be crucial to EAE development [151,152]. Further studies are necessary to determine if microglia are involved in the activation of CD8^+^ T cells found in MS lesions.

As reviewed by Piacente et al., sirtuin 1 (SIRT1) and sirtuin 2 (SIRT2), which are NAD^+^-dependent deacetylases, possibly play a role in regulating neuroinflammation and microglial activation in MS [153]. Studies of SIRT2 using LPS-induced neuroinflammation have yielded conflicting results, with one group demonstrating SIRT2^−/−^ mice experiencing an increase in pro-inflammatory cytokines and morphological changes in microglia, while another group found that administration of an SIRT2 inhibitor significantly reduced microglial activation, as well as TNF-α and IL-6 expression [153,154,155]. Current research on SIRT1 is more cohesive, however, with several groups reporting that SIRT1 may promote the transformation of activated microglia from a pro-inflammatory M1-like phenotype to a neuroprotective M2-like phenotype in EAE mice [153,156,157]. Additionally, SIRT1 may play a role in regulating inflammation, as upregulation of SIRT1 in LPS-treated microglial cell line has been shown to attenuate the expression of IL-1β and IL-6 [153,158]. These findings demonstrate that sirtuins may be a promising target for promoting resolution of neuroinflammation, particularly in MS.

### 3.2. Receptors Involved in Macrophage/Microglia Phagocytosis of Myelin Debris in MS

Myelin debris internalization by macrophages and microglia has become a well-established hallmark for MS, with studies showing that numerous receptors are involved in facilitating this event [49,159]. We therefore focus on the uptake of myelin debris in this section. The expression of CD36, a member of the class B scavenger receptor family, was elevated in murine BMDMϕ and microglia upon uptake of myelin debris in vitro [160]. CD36 expression was displayed primarily by myelin-containing phagocytes in MS lesions and spinal cord lesions from mice with EAE [160,161]. Inhibition of CD36 with sulfo-N-succinimidyl oleate (SSO) decreased the ability of macrophages and microglia to internalize myelin debris [160,162], suggesting that CD36 may be an integral receptor involved in myelin debris phagocytosis by macrophages and microglia in MS. Scavenger receptor A, a transmembrane glycoprotein that acts as a pattern recognition receptor, is constitutively expressed by macrophages and microglia and plays an important role in innate immune function [163]. It has been shown that mRNA expression of both SR-AI/II is upregulated in both infiltrated macrophages and microglia surrounding chronic active MS lesions of post-mortem human brain tissue [164]. A monoclonal antibody targeting SR-AI/II inhibited peritoneal macrophage from engulfing sciatic nerve debris in a dose-dependent manner [48,86]. The currently available research on the role of SR-AI/II highlights the mechanisms through which macrophages and microglia contribute to myelin debris clearance in the absence of opsonizing agents such as immunoglobulins or complements.

Active MS lesions in human brain tissue show significant expression of anti-human triggering receptor (TREM2) on lipid-laden macrophages surrounding the lesion areas [85,165]. Using the model of toxic demyelination induced by cuprizone (CPZ) (an animal model for MS), *Trem2^+/+^*, *Trem2^+/−^*, and *Trem2^−/−^* mice were evaluated on their microglial response to injury due to CPZ administration [85]. Corpus callosum tissue in *Trem2^−/−^* mice showed significantly increased levels of degraded myelin as compared to the corpus callosum from *Trem2^+/+^* mice, implicating that TREM2 plays a major role in microglia uptake and clearance of myelin debris [85]. A different study using EAE mice showed that blocking TREM2 activation in vivo with anti-TREM2 monoclonal antibody caused a worsened disease state, with mice exhibiting increased myelin damage associated with decreased microglial phagocytosis of myelin debris [166]. It has been shown in vitro that when apoptotic neurons are added to primary microglia culture, TREM2 knockdown microglia display significantly less phagocytosis as compared to WT microglia, again highlighting its significant role in microglia phagocytosis of myelin debris [167].

Fc receptors (FcR) may also be involved in myelin debris clearance in MS [49,50]. Active MS lesions contained both macrophages and microglia and significantly expressed FcRI, FcRII, and FcRIII [50]. Normal white matter, on the other hand, only contained macrophages that exhibited weak expression of these receptors [49,50]. In vitro experiments utilizing CSF from rabbits with EAE to pre-opsonized myelin with IgG before addition to primary rat microglia, or macrophages, showed significant increases in phagocytosis of pre-opsonized myelin by both cell types [49,168,169]. Taken together, these studies demonstrate that FcR-mediated clearance of myelin debris by macrophages and microglia plays an important role in the process of demyelination during MS.

CR3, which is expressed on both macrophages and microglia, has been shown to play a role in myelin debris phagocytosis [48,52]. A reduction in phagocytosis is observed in heat-inactivated serum, where active complement is absent, implicating that there is a complement-dependent component to myelin debris phagocytosis [48]. CR3-specific antibodies also caused reduced myelin debris phagocytosis in peritoneal macrophages, highlighting CR3’s role in macrophage uptake of myelin debris [170]. Though it is agreed upon that microglia and macrophages are able to take up myelin debris through CR3 interaction, further investigation must be conducted to fully determine the significance and implications this has in MS pathogenesis.

Mer tyrosine kinase (MerTK) and low-density lipoprotein receptor-related protein 1 (LRP1) appear to also play a role in myelin debris clearance [49]. Treatment of human monocyte-derived macrophages and microglia with MerTK antagonists caused a significant drop in myelin debris phagocytosis for both cell types [49,171]. After 4 weeks of CPZ treatment in Mertk- KO and WT mice, there was a significant reduction of microglia infiltration in the corpus callosum; however, after 7 weeks there was no significant difference between WT and KO mice. Myelin debris uptake by Mertk-KO microglia is also significantly inhibited compared to WT microglia [172]. Further research regarding MerTK is necessary to establish the significance of its impact on myelin debris phagocytosis in relation to the other mechanisms discussed in this section. Additionally, LRP1 is expressed in microglia and facilitates the uptake of myelin debris, and deletion of this gene exacerbates pathology in the EAE mice [49,173,174]. However, very little research exists on the role LRP1 plays overall in the clearance of myelin debris in MS patients and EAE mouse models. Lastly, as it pertains to microglial response in demyelinating disease, spleen tyrosine kinase (SYK) has been shown to play a neuroprotective role. EAE mice with ablated SYK exhibited increased demyelination and more aggravated paralysis than control mice. When fed a CPZ diet, mice with ablated SYK had fewer microglia present in the corpus callosum and showed impaired phagocytic ability of damaged myelin basic protein [175].

### 3.3. Implications in Pathogenesis of MS

MS is characterized by the presence of numerous autoreacting antibodies, including anti-MBP autoantibodies that contribute to the targeting and breaking down of MBP. Modulation of these autoantibodies has been shown to occur in two distinct stages: the first stage in which microglia and peripheral macrophages contribute to demyelination and neuroinflammation, and the second stage in which inflammation is eventually resolved and tissue repair in the CNS begins [176]. It has been demonstrated in vivo that during demyelination, macrophages and microglia take on a proinflammatory M1 phenotype [177,178]. M1 polarized macrophages in EAE expressed higher levels of iNOS, IL-6, IL-12, CCL2, and CXCL10 than did controls [178]. α B-crystallin (HSPB5, a molecular chaperone expressed by oligodendrocytes) treatment upregulated IL-10 expression in human microglia and macrophages [179]. Both HSPB5 and IFN-γ treatment significantly upregulated expression of proinflammatory cytokines (TNF-α, IL-12, and IL-1β) and chemokines (CXCL9, CXCL10, and CXCL11) in macrophages and microglia [179]. Microglia residing in MS lesions of brain tissue from MS patients demonstrated significant expression of IL-23, a critical component for recruitment of Th17 cells that would further contribute to neuroinflammation [180,181]. While much focus is typically paid to CD4^+^ T cells in MS pathology, as there is a strong association of MS susceptibility with MHC class II alleles, current evidence suggests there may be antigen-driven activation of CD8^+^ T cells and that they may play a significant role in MS disease progression [182]. CD8^+^ T cells are the primary T cell in the CNS of MS patients, and certain subsets of CD8^+^ T cells that can secrete IFN-γ and IL-17 have been found in perivascular spaces in active MS lesions [183]. The abundance of CD8^+^ T cells are positively correlated with intensity of axonal damage in MS [184,185]. Additionally, although both autoreactive CD4^+^ T cells and autoreactive CD8^+^ T cells are present in MS patients, those patients with relapsing–remitting MS showed a higher proportion of these autoreactive CD8^+^ T cells [186]. For complete reviews of both T cell and specifically CD8^+^ T cell involvement in MS progression see [185,187]. It is well-documented that M2 macrophages typically express high levels of anti-inflammatory cytokines such as IL-10 and TGF-β and downregulate their production of pro-inflammatory cytokines, while M2 microglia produce IL-4, IL-10, and TGF-β [188,189]. In mice brain with lysolecithin-induced focal demyelination, macrophages and microglia switched from a pro-inflammatory M1 phenotype to an anti-inflammatory M2 phenotype once remyelination began [177]. Other studies provide evidence that within MS lesions, myelin-laden macrophages, or Mye-Mϕ, expressed M2-like markers, or both M1 and M2 markers [190,191,192]. It was also noted that in vivo, M2-conditioned media increased oligodendrocyte differentiation and prevented oligodendrocyte progenitor cell apoptosis, a step that is crucial in remyelination [177,193,194]. An in vitro study showed that Mye-Mϕ inhibited lymphocyte proliferation through nitric oxide production in an antigen-independent manner [195]. These studies demonstrate that macrophages may have intermediate activation status that is dependent on spatial and temporal aspects of MS progression in the CNS. While current research points to temporally there being a tipping point at which macrophages and microglia stop provoking neuronal insult and develop more of an M2-like phenotype and begin to promote healing in MS, there is still much left to understand about this process.

Pathological patterns characterized by the presence of Mye-Mϕ, extensive IgG deposition, and Ig reactivity associated with degenerating myelin have been observed in actively demyelinating lesion areas that are consistent with T cell and macrophage-mediated inflammation [196,197]. Taking into account the numerous autoantibodies believed to be involved in MS, and the phagocytic activity of microglia and macrophages, it doesn’t seem implausible to believe there is a link between adaptive immunity and the exacerbation of MS pathogenesis by microglia and macrophages. Macrophages and microglia could be contributing to the population of anti-MBP autoantibodies that cause autoreactive tissue damage in MS patients.

In summary, the development, progression, and resolution of MS pathology depend on complex interactions between macrophages, microglia, B cells, and T cells. In demyelinating lesions, both macrophages and microglia are capable of phagocytosing myelin debris. In the brain samples from MS lesions patients, macrophages and microglia are found in close proximity to CD4^+^ T cells, which could facilitate presentation of myelin debris as antigens and assist the production of autoantibodies to myelin components. Although they are initially pro-inflammatory, both macrophages and microglia eventually take on a neuroprotective role and promote CNS recovery. Further research is needed to understand this process. Studies in EAE mice have shown attenuation of disease progression when macrophages or microglia are incapable of presenting antigens. Interestingly, when only microglia are incapable of presenting antigen and macrophages are unaffected, macrophage recruitment in the CNS and disease progression are attenuated. Therefore, microglia may play a more significant role in modulating CNS damage than macrophages. Further investigation into therapeutics that can skew microglia towards an M2 phenotype may provide additional downstream benefits in MS treatment.

## 4. Alzheimer’s Disease

Alzheimer’s disease (AD), the most common form of dementia, is a progressive neurodegenerative disease that causes progressive impairment of both behavioral and cognitive functions in patients [198]. AD is responsible for 60–70% of the more than 55 million dementia cases worldwide [199]. The two hallmark neuropathological signs associated with AD are senile plaques and neurofibrillary tangles (NFTs). Senile plaques are primarily composed of aggregated extracellular amyloid-β (Aβ) proteins, which can exert a neurotoxic effect that can stimulate damage to axons and dendrites, as well as cause synaptic loss [200,201]. NFTs are primarily composed of the hyperphosphorylated form of microtubule-associated protein tau that aggregates in neurons, ultimately contributing to neuronal death [201,202,203]. Microglia have been shown to make a considerable contribution to the Aβ and tau pathologies seen in AD through various pathways [204]. Early symptoms of AD typically include impaired memory, but then can progress into severe cognitive and behavioral impairment with symptoms such as anxiety, paranoia, insomnia, and irritability [198].

### 4.1. Immune Responses and Antigen Presentation in AD

Numerous studies have been performed to assess the potential of Aβ autoantibodies to be used as a diagnostic marker for AD, with mixed results. When unbound Aβ autoantibodies were assayed in serum from AD patients, they were found in lower levels than those found in sera from control populations [53,54,55,56]. Similarly, the levels of Aβ autoantibodies in the CSF of AD patients were lower than those of healthy controls [205,206,207,208]. In contrast to these findings, when sera from patients was purified to isolate only IgG bound to Aβ42, AD patients showed four-fold higher titers when compared to controls [209]. This same study also found that Aβ IgG titers were negatively correlated with the cognitive status of the patients [209]. Furthermore, levels of antigen-bound Aβ auto-antibody in CSF were higher in AD patients, as well as negatively correlated to cognitive status across groups [210]. The presence of autoantibodies against tau protein, a protein mainly involved in maintaining microtubule stability in neuronal axons, has been detected in healthy adults, AD diagnosed adults, and in children [211,212]. The elevated levels of phosphorylated and non-phosphorylated tau protein were detected in the CSF of AD patients compared to healthy controls [208,213,214,215,216]. Autoantibodies against the β subunit of adenosine triphosphate (ATP) synthase have been identified in the serum of AD patients, but unlike autoantibodies to tau and Aβ, they were not identified in healthy controls [217,218]. This may be the basis for the mitochondrial hypothesis of AD, which was formed based on findings that mitochondrial infrastructures and lower glucose utilization is observed in the brains of AD patients, implying that mitochondria may play a role in disease progression [218,219]. Through the use of the 5xFAD transgenic mouse model of AD, some studies have shown elevated levels of anti-ceramide antibodies compared to WT mice; however, this does not appear to have been studied in humans [56,220]. As reviewed by Wu and Li, there are numerous other autoantibodies under investigation for their potential as biomarkers and therapeutic targets for AD [56]. Autoantibodies related to various neurotransmitters and receptors, glial markers, lipids, cellular enzymes, and vascular materials are all possible targets based on their implicated involvement in AD pathogenesis. With many of these targets, it is still unknown if they are pathogenic, or protective, in regard to their involvement in AD.

It has been fairly well established that microglia in the brain of AD patients express the MHC II molecule HLA-DR, and MHC II expressing microglia are clustered around Aβ plaques [57,58,59,60]. An MHC II costimulatory molecule, CD40, is expressed on microglia in vitro and in vivo [221,222,223]. Tau overexpression in rats induced upregulation of TNF-α at the mRNA level, while Aβ treatment upregulated both TNF-α and CD40 expression in human microglia [224,225]. Upregulation of TNF-α in this scenario is of importance since autocrine signaling of TNF-α is crucial for IFN-γ induced-CD40 expression [226,227]. Expression of MHC I molecule has also been observed in microglia from AD patients’ brains but at relatively low levels [228]. Though it is undetermined if microglia are mounting processed antigen onto MHC I or MHC II molecules, current findings indicate that they at least have the potential to do so in AD.

While MHC I and II presentation of antigens by microglia in AD has yet to be shown, there have been various studies in AD investigating the presence of the CD8^+^ T cells, the cells to which microglia would potentially present antigen. Multiple groups have shown in both mouse models and samples from AD patients that CD8^+^ T cells are present in increased levels in the CSF and blood and that they migrate to the brain from the periphery [229,230,231,232,233]. Surveys of CD4^+^ T cells have led to controversial findings, with numerous groups reporting decreased CD4^+^ T cell levels in AD patients, while at least one group has reported an increase in CD4^+^ T cells [233,234,235,236]. A case study of AD patient CSF samples revealed that levels of peripheral Th17 cells were elevated compared to controls, and the proportion of regulatory T cells (Treg) was positively correlated with the level of pTau181 and total Tau in AD patients, implying that these T cells may be associated with neurodegeneration in AD [237,238]. However, there was not a significant difference between Treg populations in the CSF of AD patients and controls [237,238]. There were, however, limitations to this study: with difficulty obtaining CSF from healthy control samples, patients without cognitive impairment were used as controls [237,238]. It has been noted that there is an increase in Th1 and Th17 cell response in AD patients’ brains, while Th2 cell presence is substantially lower in comparison [239]. As reviewed by Sabatino et al., evidence showing a pathogenic role of B cells in AD is limited [239]. Activated B cells in the AD mouse model were elevated compared to controls [240]. However, AD patients display lower levels of peripheral B cells, and studies on B cells in AD mouse models do not agree on whether they are deleterious or protective [239].

### 4.2. Receptors Involved in Macrophage/Microglial Phagocytosis of Aβ and Tau in AD

It has been demonstrated that fibrillar Aβ (fAβ) is capable of activating microglia and, in turn, stimulating an increase in phagocytic activity [241,242,243]. There is a multicomponent fAβ receptor complex consisting of CD36, α6β1, and CD47 that facilitates the adhesion of fAβ to microglia [242]. BV-2 cells, a microglia cell line, are capable of phagocytosing fAβ and trafficking it into phagosomes after uptake [244]. Incubation of BV-2 cells with unique agonists to each individual component of the CD36, α6β1, and CD47 prevented the initiation of phagocytosis, implying that each receptor in the complex is crucial for microglia uptake of fAβ [244]. Scavenger receptor SCARA-1 has been shown to significantly increase phagocytosis of soluble Aβ by primary microglia, with SCARA-1 null mice showing increased accumulation of Aβ in brain tissue [61,62,63]. The role of CD36 has also been investigated in microglia phagocytosis of Aβ and was found to possibly be effective only in the process of phagocytosing fAβ and not soluble Aβ [62].

Numerous groups have also shown that Toll-like receptor (TLR)-2 (TLR2), TLR4, and co-receptor CD14 play a role in fAβ binding and microglia activation [245,246,247]. Microglia isolated from WT mice phagocytosed significantly more fAβ than did CD14-KO microglia, while there is no difference in the uptake of control particles such as polystyrene microspheres between WT and CD14-KO microglia [247]. Brains of AD patients showed strong expression of CD14 on microglia, while control subjects showed no parenchymal expression [247]. The genetic depletion of CD14, TLR2, and TLR4 or function-blocking antibodies against CD14, TLR2, and TLR4 significantly attenuate the phagocytic capacity of BV-2 cells for fAβ, highlighting their importance in microglial phagocytosis of fAβ [245]. Furthermore, CD14, TLR2, and TLR4-deficient microglia exhibited reduced activation of MAP kinase p38 upon exposure to fAβ, inhibiting ROS production and phagocytosis [245]. The GTPase Rac, whose activation by GTP-loading and cytosol to membrane translocation is regulated by p38 activity, is believed to play a role in phagocytosis [245]. Treatment of microglia with p38 inhibitor, SB203580, before exposure to fAβ reduced fAβ-stimulated phagocytosis, providing evidence that GTPase Rac plays a role in microglia phagocytosis of fAβ in AD [245].

TREM2 has been linked to the promotion of Aβ phagocytosis [248,249,250]. Aβ treatment for up to 24 h increased microglia phagocytic response; however, cells silenced for TREM2 had phagocytosis restored to the level of control cells [250]. There is a decrease in Aβ phagocytosis in TREM2-KO mice compared to WT mice, and TREM2-KO mice have a marked decrease in CD36 mRNA levels, implying a link between TREM2 and CD36 in the phagocytosis of Aβ [248].

The G protein-coupled receptor (GPCR) formyl peptide receptor 2 (FPR2), whose human homolog FPR-like-1 (FPRL-1), is normally expressed constitutively at low levels but is upregulated when stimulated with TNF-α, is expressed by both microglia and astrocytes upon activation by Aβ [61,251,252]. FPRL-1 has been shown to act as a receptor for Aβ and colocalized with Aβ in the cytoplasm of monocytes-derived macrophages [61,253,254]. Another GPCR, chemokine-like receptor 1 (CMKLR1), is colocalized with internalized Aβ42 in microglia, suggesting CMKLR1 may play a role in Aβ processing and clearance [61,255]. It has been shown that SYK protein is responsible for intracellular regulation of microglia activity in 5xFAD mice. Microglia in mice with loss of SYK expression exhibited impaired activation, proliferation, and phagocytosis of Aβ [175]. Lastly, microglia are also capable of internalizing soluble and insoluble tau protein isolated from AD patient brain tissue in vivo and in vitro [64,65]. Primary microglia isolated from CX3CR1 KO mice have significantly lower levels of tau uptake when compared to cells from WT mice. Stereotaxic injection of tau into WT and CX3CR1 KO mice demonstrated attenuation of tau internalization in the KO mice, similar to the in vitro experiments, implying that CX3CR1 plays a role in the phagocytosis and clearance of extracellular tau in AD [256,257].

### 4.3. Implications in Pathogenesis of AD

In the early stages of AD, microglia are central to the recognition and clearance of Aβ plaques in the brain, being shown to slow down disease progression in mouse models of AD [66,258,259,260]. However, Aβ also activates microglia, which contributes to Aβ plaque formation. Microglial uptake of Aβ in 5xFAD mice ultimately leads them to undergo cell death, releasing accumulated Aβ back into the environment and further contributing to Aβ plaque growth [261,262]. Studies have demonstrated a feedback loop in which apoptosis-associated spec-like protein containing a CARD (ASC) forms composites with Aβ, which activates the NOD-like receptor protein (NLRP3) inflammasome [262,263]. Upon activation of NLRP3, Aβ uptake and clearance is reduced, and the microglia undergo pyroptotic cell death [262,263]. Upon death, microglia release ASC back into the environment to further perpetuate the cycle [262,263]. Further study is warranted to properly determine if microglia have an overall net positive or negative contribution to Aβ plaque formation in AD.

Microglia produce a wide array of inflammatory cytokines and chemokines that are elevated in AD. Microglia appear to be skewed towards an M1 pro-inflammatory phenotype upon activation by Aβ, promoting the production of NO, ROS, TNF-α, and up-regulating genes for pro-inflammatory cytokines and chemokines such as IL1A, IL1B, IL6, IL8, CCL2 [264,265,266,267,268]. TNF-α, as well as C1q and C3, have been linked to microglia-mediated neuronal loss in AD mouse models [268,269,270]. Overexpression of the receptor for advanced glycation end products (RAGE) has been linked to increased IL-1β and TNF-α production upon stimulation in AD mouse models [271,272]. Knockdown of RAGE via siRNA transfection significantly reduced Aβ levels when compared to controls, possibly through changes in the regulation of β- and γ-secretase activity [271,273]. These findings lend more credence to the theory that microglia in early-stage AD fit the M2-neuroprotective phenotype, but with increases in Aβ plaques in the brain and continued interaction with Aβ-induced pro-inflammatory cytokine production, they switch to an M1-neurotoxic phenotype [264,274,275].

Studies investigating the reactivity and proliferation of peripheral T lymphocytes from AD patients have yielded conflicting results. One study measured no significant difference in T cell reactivity to Aβ42 protein and found a significantly decreased response to human mitochondrial peptides when compared to healthy controls [276]. A recent study detected T cell reactivity to Aβ, tau, and amyloid precursor protein (APP); however, there was no significant difference between AD patients and healthy controls [277]. Another study corroborated these results, finding that various Aβ did not significantly stimulate proliferation of T cells from peripheral blood [278]. When comparing the T cell reactivity to Aβ42 between AD patients, healthy middle-aged and healthy elderly subjects, reactivity was not significantly different between elderly controls and AD patients [279]. Although T cell reactivity between elderly controls and AD patients was not significantly different, both groups were significantly elevated when compared to middle-aged controls [279]. Additional research should be conducted to definitively confirm if T cell reactivity is dependent on secondary factors besides AD pathogenesis.

Immunoglobulins have been shown to associate with neurons in the brains of AD patients, typically in neurons that were undergoing degeneration [280,281,282]. One study found that most neurons that colocalized with Aβ also showed IgG expression [282,283]. There is research that implies Aβ autoantibodies may play a protective role against AD pathogenesis, with one study finding that brain tissue from AD patients that had high amounts of IgG-decorated Aβ plaques had drastically reduced Aβ plaque burdens compared to brain tissue from AD patients exhibiting lower amounts of IgG-labeled plaques [282,284]. Patients with increased IgG-labeled plaques also exhibited a greater number of phagocytic microglia than those with decreased IgG-labeled plaques [282,284]. Further research is necessary to determine if autoantibodies present in AD patients play an arresting or exacerbating role in disease progression.

In summation, many avenues of investigation into the possible role of microglia in antigen presentation in AD have yielded conflicting results. Microglia have been shown to phagocytose various forms of Aβ and tau protein using a variety of receptors. Microglia in the brains of AD patients express MHC I and II, and MHC II expressing microglia have been observed to cluster around Aβ plaques. While studies suggest elevated CD8^+^ T cell levels in AD patients, the findings on CD4^+^ T cell levels are conflicting. Current research shows that autoantibody levels are elevated in AD patients, indicating that microglia may initiate and assist autoantibody production through antigen presentation. Unlike in cases of neurotrauma and MS, the role of microglia in AD remains unclear. Investigation into methods to manipulate microglia to have a net positive effect on Aβ and tau clearance, or to prevent their shift to a pro-inflammatory phenotype as disease progression, could prove beneficial for AD treatment.

## 5. Parkinson’s Disease

Parkinson’s disease (PD), a chronic progressive neurodegenerative disorder caused by loss of dopaminergic neurons in the substantia nigra (SN), affects over 8.5 million people globally [285,286,287,288]. α-synuclein (αS) is an abundantly expressed protein in the nervous system that is believed to act as a modulator of neurotransmitter release [289,290]. Under physiological conditions, extracellular unfolded αS monomers are in equilibrium with membrane-bound monomers [289,290]. These unfolded monomers are able to bind to lipid membranes, as well as form tetramers that do not abnormally aggregate. As seen in PD, these unfolded monomers can form oligomers, which will eventually form fibrils (dubbed “on-pathway” oligomers), while others will form oligomers that are unable to form fibrils (dubbed “off-pathway” oligomers) [289,290]. These oligomers can lead to neurotoxic effects by inducing mitochondrial dysfunction, endoplasmic reticulum stress, proteostasis disruption, synaptic impairment, cell apoptosis, and inflammation [289]. Microglia, in concert with T cells, play a central role in mediating the aforementioned neurotoxic effects in the CNS [291]. Individuals affected by PD exhibit motor symptoms such as bradykinesia, muscle tone rigidity, resting tremor, and postural instability as well as non-motor symptoms such as sleep disorders, sensory abnormalities, and autonomic dysfunctions [292,293].

### 5.1. Immune Response and Antigen Presentation in PD

Post-mortem brain tissue samples from patients with PD showed colocalized expression of IgG antibodies with αS in SN neurons [294]. Autoantibodies generated against αS have been detected in both the cerebrospinal fluid (CSF) and blood sera from PD patients [295]. There is disagreement as to if levels of autoantibodies against αS are elevated in individuals with PD. When blood serum samples from patients with sporadic PD or familial PD were compared to that of healthy patients, a positive correlation was found only for patients who had familial PD and not for those who presented with sporadic PD [296]. In another study that excluded patients who had familial, atypical, or secondary parkinsonism, the median detected level of autoantibodies to αS were higher in sera of PD patients compared to sera of the control group [297]. On the other hand, several groups have shown that αS autoantibody levels in PD patients’ sera are not significantly elevated as compared to healthy control groups, and, in some instances, levels in PD patients are actually lower than in controls [295,298,299]. The conflicting findings regarding autoantibodies are the same in the CSF of PD patients as they are in blood sera [300]. While some groups have found autoantibodies to αS to be higher in the CSF of PD patients as compared to CSF from controls regardless of the severity of PD [295,301], other groups have demonstrated that there is no significant difference between the two groups [302]. Autoantibodies reactive to GM-1 ganglioside and neuromelanin (NM) are present in higher levels in PD patients as compared to controls but results so far are limited [67,303,304]. Differences in results between groups could be explained by clinical heterogeneity of patients amongst studies or possibly due to the variability of assays used [300].

Important to the cell-mediated immune response, MHC II expression was found to be increased on myeloid cells coinciding with increased peripheral CD4^+^ and CD8^+^ T cell infiltration into the CNS in PD patients, due in part to the observed increase in permeability of the blood–brain barrier [305,306,307,308,309,310]. This increase in MHC II expression was dependent on the level of microglia activation in the brain [305,306,307,308,309,310]. Analysis of peripheral CD4^+^ T cell subsets in PD patients has shown proportions of Th1/Th2 and Th17/Treg are significantly higher in favor of Th1 and Th17 cells [311]. Peripheral B cell populations in PD patients are found to shift from naïve to unswitched memory B cells, which could potentially contribute to T cell-dependent production of antibodies that infiltrate the brain in PD [312].

### 5.2. Receptors Involved in Macrophage/Microglia Phagocytosis of αS and NM in PD

Microglia have the ability to phagocytose numerous materials associated with PD, including αS and NM, with multiple types of receptors potentially playing a role in their uptake. Internalization of αS aggregates by primary microglia has been shown to occur in vitro [66,68,69]. TLR4 is crucial for microglia activation and subsequent uptake of αS. WT primary murine microglia demonstrated increased phagocytic activity upon treatment with either soluble, fibrillary, or truncated αS, while microglia isolated from TLR4-deficient mice demonstrated reduced phagocytosis after the same treatment [68]. Further research has linked TLR4-mediated engulfment of αS to a process dubbed “synucleinphagy”. In this process, αS interaction with TLR4 triggers NF-κB signaling to upregulate transcription of p62, an autophagy receptor that is necessary for αS degradation via autophagy [313]. TLR2 can interact with oligomeric (but not monomeric or fibrillar) αS, leading to its internalization by microglia [314,315]. It has been shown that CR3 and CR4, (also known as p150,95, integrin αXβ2, or CD11c/CD18) bind to oligomeric and fibrillar forms of αS, in turn leading to their phagocytosis [316,317]. Interactions between αS and CD11b expressed on microglia have been observed; however, further investigation is warranted to show internalization of αS due to CD11b interaction [318].

Both in vitro and post-mortem observations have shown that microglia are capable of phagocytosing NM [70,71,72]. One group investigated the potential of CR3 involvement in phagocytosis, noting that NM phagocytosis was attenuated in microglia depleted for CR3 as compared to WT microglia [72]. Complement factor C1q may also play a role in microglia phagocytosis by opsonizing extracellular NM. Microglia express surface receptor C1qRp, which interacts with C1q to enhance Fc receptor and CR1-mediated phagocytosis [319]. Brain tissue from patients with PD showed increased levels of C1q^+^ microglia with more proliferative activity located near NM^+^ neurons [320]. The same study showed that significantly more C1q^+^ microglia in the SN pars compacta (SNpc) of PD patients contained NM granules in their cytoplasm as compared with controls [320].

Post-mortem brain tissue from PD patients showed significant amounts of SN microglia with high levels of HLA-DR expression, an MHC II molecule that is a crucial component of antigen presentation to CD4^+^ cells [58]. Overexpression of αS leads to an upregulation of MHC II in microglia [321,322]. Further study of post-mortem brain tissue from PD patients showed IgG colocalized with αS in pigmented SN neurons, which exhibited a positive correlation with MHC II expressing microglia [294]. While all of the signs of potential antigen presentation by microglia are present in PD, further research is still necessary to conclusively demonstrate that microglia are in fact presenting antigens in PD.

### 5.3. Implications in PD Pathogenesis

In two studies utilizing αS overexpression, one model utilizing MHC II KO mice and the other utilizing FcγR^−/−^ mice, the mice experienced little to no dopaminergic cell loss in the SNpc and the SN, respectively [322,323]. These findings implicate the involvement of microglia in the mediation of dopaminergic cell loss. NM also appears to induce microglia-mediated neurotoxicity. Microglia activated by NM in mixed neuron–glia cocultures caused a decrease in the number of tyrosine-hydroxylase (TH) neurons. In vitro experiments were performed using embryonically derived primary mesencephalic neuron–glia cultures from phagocytic oxidase (PHOX) KO rats. Without PHOX, a subunit of NADPH oxidase that is responsible for the production of superoxide and H_2_O_2,_ ROS production is inhibited. After exposure of the neuron–glia cultures to NM to induce microglial activation, there was a greater number of TH neurons in the PHOX^−/−^ derived culture than in the PHOX^+/+^ derived culture, implying that microglial ROS production may play a significant role in NM-stimulated microglia-mediated neurotoxicity [72]. These results also held true in adult rats that received a stereotaxic injection of NM in the SN portion of their brains, where they found significant activation of microglia that induced degeneration of dopaminergic neurons [72]. This increase in ROS production is also demonstrated by BV-2 cells after 24 h exposure to NM [324]. NM treatment also significantly increased mRNA expression of proinflammatory mediators such as TNF-α, IL-1β, IL-6, and iNOS in BV-2 cells [324].

When studying the effect of CD8^+^ and CD4^+^ T cell involvement in 1-methyl-4-phenyl-1,2,3,6-tetrahydropyridine (MPTP) mouse models of PD, a notable decrease in dopaminergic cell death was observed in *Cd4^−/−^* mice compared to WT littermates and *Cd8a^−/−^* mice [310]. Co-cultures of induced pluripotent stem cell (iPSC)-derived midbrain neurons (MBNs) and T lymphocytes from sporadic PD patients led to increased cell death of the iPSC-derived MBNs, suggesting that these T cells may be neurotoxic and contribute to neuronal death in PD [325,326]. In a study assessing how IgG from PD patient sera affects the SNpc, purified IgG was injected into the SN of rats, which leads to a significant reduction of SNpc neurons in rats receiving purified IgG from PD patients compared to rats receiving purified IgG from non-disease controls [326,327].

In summary, it appears probable that microglia are presenting antigen to T cells and are certainly mediating neuronal damage in PD patients. Microglia have been shown to internalize αS through several receptors and are capable of internalizing NM. MHC II expression has been observed on microglia in PD patients, indicating the possibility of presenting internalized αS or NM to T cells present in the brain. While higher levels of autoantibodies have been detected in PD patients compared to controls, studies have shown conflicting findings. In addition, decreases in dopaminergic cell loss is observed in MHC II^−/−^ or CD4^−/−^ mice, and injection of purified IgG from PD patients into the SN of rats results in significant reduction of SNpc neurons. Taken together, these findings suggest that microglia are exacerbating the pathogenesis of PD, both directly through microglia-mediated neurotoxicity and indirectly through T cell interactions.

## 6. Conclusions

Macrophages/microglia are a multifunctional cell type that play multiple roles in disease progression within the CNS. A complete understanding of the mechanisms and initiating factors that produce their beneficial or detrimental contributions to CNS diseases is a major ongoing focus for neuropathology researchers. Within this review, we discussed neurotrauma and the most common neurodegenerative diseases and presented evidence for how macrophages/microglia may be impacting the progression of these disorders. Macrophages and microglia play a critical role in neuroinflammation in various CNS disorders and are responsible for clearance of cell debris and proteins such as Aβ, tau, and αS at the lesion sites. Furthermore, macrophages and microglia can present peptide antigens derived from engulfed cell debris and proteins and thus act as the professional antigen presenting cells in CNS lesions and hence result in activation of effector lymphocytes and initiation of specific immune response. The process of macrophage/microglia initiated adaptive immune response could be an additional mechanism for their contribution to disease progression. Gaining insight into the multifaceted roles of macrophages and microglia in CNS diseases is of critical importance in not only ultimately reaching the point of having a thorough and complete understanding of CNS disease pathogenesis but also in generating novel therapeutic strategies.

## Figures and Tables

**Figure 1 ijms-24-05925-f001:**
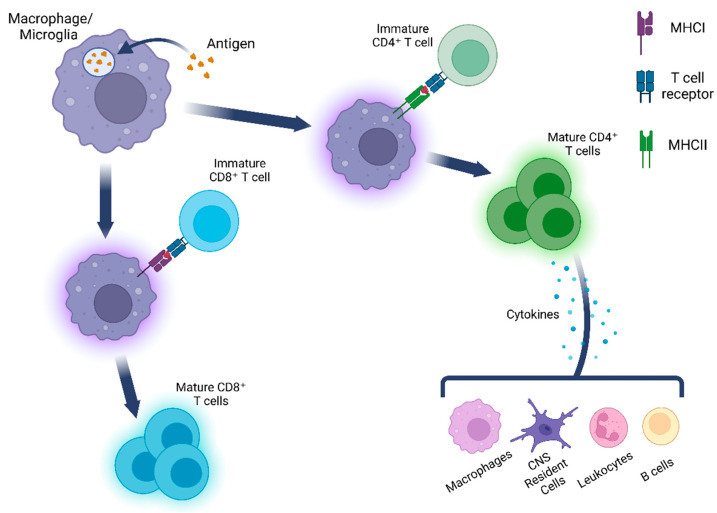
Macrophage and microglia phagocytose materials that subsequently present antigens to both CD4^+^ and CD8^+^ T cells. It is these CD8^+^ T cells that are then able to activate various cell types such as macrophages and B cells and generate a potent immune response.

**Table 1 ijms-24-05925-t001:** Overview of CNS diseases and injuries discussed in this review.

CNS Disease/Injury	Macrophages/Microglia Presentation to APC	Presence of Autoantibodies	Major Autoantibody-Antigens *	References
Spinal Cord Injury	Microglia and infiltrating macrophages demonstrate phagocytic capability, as well as display MHC I and II in vitro	√	GM1 **, GFAP, CRMP2, MBP	[21,25,26,27,28,29,31,37,38,39,40,41,42]
Traumatic Brain Injury	Microglia and infiltrating macrophages demonstrate phagocytic activity, but MHC I and II expression appears dependent on specific factors	√	β-tubulin class III, GFAP, S100B, MBP	[32,43,44,45,46,47]
Multiple Sclerosis	Microglia and macrophages phagocytose myelin via FC receptor, CR3 receptor	√	MBP, MOG **	[48,49,50,51,52]
Alzheimer’s Disease	Microglia phagocytose Aβ and tau and express MHC II	√	Aβ, tau, beta subunit of ATP	[53,54,55,56,57,58,59,60,61,62,63,64,65]
Parkinson’s Disease	Microglia internalize αS aggregates and NM, and are positive for MHC II	√	αS, GM-1 ganglioside, NM	[58,66,67,68,69,70,71,72]

αS: alpha-synuclein, NM: neuromelanin, MBP: myelin basic protein, MOG: myelin oligodendrocyte glycoprotein, GFAP: glial fibrillary acidic protein. * The antigens presented here are the mostly commonly detected or studied and are not a comprehensive list of autoantibody antigens discovered in their respective disease or disorder; ** Conflicting findings regarding these results have been demonstrated and require further investigation.

## Data Availability

Not applicable.

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
