# Peer review of "Immune Regulatory Functions of Macrophages and Microglia in Central Nervous System Diseases"

_ijms, 2023, doi:10.3390/ijms24065925_

Round 1

Reviewer 1 Report

In the manuscript of Poppel and colleagues the immunoregulatory role of macrophages and microglia in the central nervous system in diseases has been studied. In particular, autoantibody antigens and their role in neurotrauma and neurodegenerative diseases have been the focus of this review and the function of receptors and microglia in uptaking cell debris after injury leading to their activation and subsequent antigen presentation has been studied.

Chapter and subchapters are numbered incorrectly. Please revise.

The review is well written and it contains all the information necessary for the reader to understand this process.

I would suggest the authors to add 1or 2 figures explaining the information they gave in the manuscript and the events described. 

Author Response

We thank the editor, guest editors, and reviewers for taking the time and effort to review our manuscript and provide us with valuable feedback. We have addressed all the queries raised and made the necessary revisions using track changes. Furthermore, we have highlighted the changes made upon the reviewers' request.

Chapter and subchapters are numbered incorrectly. Please revise.

- The chapters and subchapters have been revised.

The review is well written and it contains all the information necessary for the reader to understand this process.

- We appreciate the reviewer’s positive comments on the manuscript.

I would suggest the authors to add 1or 2 figures explaining the information they gave in the manuscript and the events described.

- We have added a figure to the manuscript to illustrate the process by which macrophages/microglia activate numerous cell types after antigen presentation.

Reviewer 2 Report

This is an interesting work but major information is missing to conclude.

The title of the review “Immune regulatory functions of macrophages and microglia in central nervous system diseases” doesn’t mention the specificity of the review which is the roles of the macrophages and microglia in autoimmune disorders.

In the abstract, I think you have to change:

 “(1) their involvement in the uptake and processing of materials during the disease progression, (2) the processes by which they may be presenting antigen, (3) the types of immune responses that have been shown to be induced in the specific disease state, and (4) what implications on the pathogenesis of the diseases they may have.   “  

By (1) the types of Immune responses and the processes of antigen presentation, (2) receptors involved in Macrophage/Microglial Phagocytosis of myelin debris, and  (3) Implications on the Pathogenesis of the diseases

 You have to review the numbers of the succession of titles of all the article

In the introduction of CNS diseases and injuries discussed in this review, you have to keep the same logic: lines 49, 194, 410 and 587.

- definition of the disease

- the number of individuals impacted worldwide and in the same year. 

- causes

- symptoms

The review is very rich in information and studies but the author has to make it more structured in the succession of information, especially you have to valorize the points of difference between macrophages and microglia and discuss these differences. 

At the end of each disease treated in this review, the author should make a recap to show his contribution and discussion in the article.

The author has to reinforce the review with figures and tables.

The author has to reinforce the review with the molecular mechanism.

You have to show the major revisions in the text, with a different color text, by highlighting the changes.

All these remarks must be taken into consideration. Answers to these criticisms will reinforce the quality of the manuscript and will permit us to conclude with more accuracy.

Author Response

We thank the editor, guest editors, and reviewers for taking the time and effort to review our manuscript and provide us with valuable feedback. We have addressed all the queries raised and made the necessary revisions using track changes. Furthermore, we have highlighted the changes made upon the reviewers' request.

The title of the review “Immune regulatory functions of macrophages and microglia in central nervous system diseases” doesn’t mention the specificity of the review which is the roles of the macrophages and microglia in autoimmune disorders.

 - While it is true that the title of our review, "Immune regulatory functions of macrophages and microglia in central nervous system diseases," does not explicitly mention the roles of macrophages and microglia in autoimmune disorders, we believe it covers a range of CNS diseases, including MS. Therefore, we still believe that the title accurately reflects the content of the review.

 In the abstract, I think you have to change:

 “(1) their involvement in the uptake and processing of materials during the disease progression, (2) the processes by which they may be presenting antigen, (3) the types of immune responses that have been shown to be induced in the specific disease state, and (4) what implications on the pathogenesis of the diseases they may have.   “ 

By (1) the types of Immune responses and the processes of antigen presentation, (2) receptors involved in Macrophage/Microglial Phagocytosis of myelin debris, and  (3) Implications on the Pathogenesis of the diseases

- We have updated the abstract to reflect your advice with the exception of the words “myelin debris”. This term may be too specific as our manuscript discusses a range of materials being phagocytosed by macrophages and microglia in different diseases. Therefore we use a more general term such as cell debris or molecules that encompass all materials discussed in the manuscript.

You have to review the numbers of the succession of titles of all the article

- We have revised this in the master version of the manuscript.

In the introduction of CNS diseases and injuries discussed in this review, you have to keep the same logic: lines 49, 194, 410 and 587.

- definition of the disease

- the number of individuals impacted worldwide and in the same year.

- causes

- symptoms

 - We have updated each of the introduction paragraphs to follow this outline and maintain consistency between each disease type. 

The review is very rich in information and studies but the author has to make it more structured in the succession of information, especially you have to valorize the points of difference between macrophages and microglia and discuss these differences.

- We appreciate your feedback on the structure of our review. In response, we have made a point to highlight and add information regarding whether macrophages and/or microglia are understood to be present and contributing to the disease progression that we discuss in each disease type. We agree that the topic of dissecting the specific differing roles of macrophages versus microglia is an interesting one and could serve as a separate review article, which we are interested in exploring in the future.

At the end of each disease treated in this review, the author should make a recap to show his contribution and discussion in the article.

- This is an excellent suggestion. We have now added a short conclusionary paragraph to the end of each section recapping the information discussed in the section as well as discussing its importance.

The author has to reinforce the review with figures and tables.

- We have added a figure to the manuscript to illustrate the process by which macrophages/microglia activate numerous cell types after antigen presentation.

The author has to reinforce the review with the molecular mechanism.

 - With this suggestion in mind, we have added specific details regarding the molecular mechanism of several processes including the phagocytosis pathway, as well as how cytokines like TNF-α contribute to inflammation through various intracellular pathways.

You have to show the major revisions in the text, with a different color text, by highlighting the changes.

All these remarks must be taken into consideration. Answers to these criticisms will reinforce the quality of the manuscript and will permit us to conclude with more accuracy.

Reviewer 3 Report

The review by Poppel et al. summarises the findings about macrophages and microglia, which suggest their role in triggering the adaptive immune response in various disorders affecting the central nervous system. The authors introduce the multiple roles of macrophages and microglia in CNS pathologies; then, they concentrate on the role that antigens presentation and autoantibodies production can have in the pathogenesis of each disease. The authors address neurotrauma, multiple sclerosis, Alzheimer’s disease, and Parkinson’s disease, providing a brief introduction listing symptoms and the expression profile of immune cells for each disease. The authors then analyze the immune response activated by macrophages and microglia, the receptors involved in the phagocytosis of disease-specific hallmarks (e.g., myelin for MS, α-synuclein for PD), and the involvement in the disorder’s pathogenesis. The authors conclude by highlighting and justifying the importance of further studies, unraveling the role of macrophages and microglia in CNS disorders.

Since microglia participate in the modulation of neuronal function by regulating the elimination (pruning) of weaker synapses in both physiologic and pathologic processes, and impairments in synaptic pruning disrupt the excitatory versus inhibitory balance (E/I balance) of synapses, the analysis of the autism spectrum disorder (ASD), Schizophrenia and Epilepsy should be added.  Andoh et al. J. Clin. Med. 2019, Koyama et al. Neuroscience Research 2015, Sellgren et al. Nat Neurosci 2019

Moreover, information related to the role of sirtuins in neuroinflammation should be added, with some missing references: Chen et al. J. Neurosci. 2017, Parodi et al. Neuropathol. 2015, Pais et al. EMBO J. 2013, Wang et al. Neurochem. Res. 2016, Li et al. J. Mol. Neurosci. 2015, Giacometti et al. Oxidative Med. Cell. Longev. 2020, Piacente et al. Int. .J Mol. Sci. 2022, Ferrara et al. J. Neuroinflamm. 2020

It would greatly help the reader visualize a schematic depiction of the involvement of macrophages/microglia in the phagocytosis of the hallmarks of each disease, with the major players involved.

The text is clear and well-structured, with some minor mistakes:

-          line 43: studied;

-          line 90: major histocompatibility complex;

-          line 469: T cells (Treg) cells;

-          the sentence in lines 239-243 is precisely the same as 371-373;

-          paragraphs’ sequential numbering to be fixed.

Author Response

We thank the editor, guest editors, and reviewers for taking the time and effort to review our manuscript and provide us with valuable feedback. We have addressed all the queries raised and made the necessary revisions using track changes. Furthermore, we have highlighted the changes made upon the reviewers' request.

Since microglia participate in the modulation of neuronal function by regulating the elimination (pruning) of weaker synapses in both physiologic and pathologic processes, and impairments in synaptic pruning disrupt the excitatory versus inhibitory balance (E/I balance) of synapses, the analysis of the autism spectrum disorder (ASD), Schizophrenia and Epilepsy should be added.  Andoh et al. J. Clin. Med. 2019, Koyama et al. Neuroscience Research 2015, Sellgren et al. Nat Neurosci 2019.

- Thank you for your feedback, we appreciate your input. After considering the inclusion of synaptic pruning in our current review and the length of our manuscript, we have decided that this process deserves its own review article. We agree that it is an interesting and important topic, and we would like to produce a separate review article that specifically discusses the role of synaptic pruning in ASD, Schizophrenia, Epilepsy, and Downs Syndrome. We will explore this topic in more depth and provide a comprehensive review in the future. Thank you for your suggestion.

Moreover, information related to the role of sirtuins in neuroinflammation should be added, with some missing references: Chen et al. J. Neurosci. 2017, Parodi et al. Neuropathol. 2015, Pais et al. EMBO J. 2013, Wang et al. Neurochem. Res. 2016, Li et al. J. Mol. Neurosci. 2015, Giacometti et al. Oxidative Med. Cell. Longev. 2020, Piacente et al. Int. .J Mol. Sci. 2022, Ferrara et al. J. Neuroinflamm. 2020

- Thank you so much for this suggestion. After reviewing papers related to sirtuins and references provided, we have added a paragraph in Section 2.1 discussing the role of sirtuins in the context of multiple sclerosis, as well as them being a potential focus for research into the resolution of neuroinflammation. 

It would greatly help the reader visualize a schematic depiction of the involvement of macrophages/microglia in the phagocytosis of the hallmarks of each disease, with the major players involved.

- We have added a figure to the manuscript to illustrate the process by which macrophages/microglia activate numerous cell types after antigen presentation.

The text is clear and well-structured, with some minor mistakes:

-          line 43: studied;

-          line 90: major histocompatibility complex;

-          line 469: T cells (Treg) cells;

-          the sentence in lines 239-243 is precisely the same as 371-373;

-          paragraphs’ sequential numbering to be fixed.

- Thank you for these detailed suggestions. We have gone through the manuscript and corrected these issues.

Round 2

Reviewer 2 Report

In this version of the article “Immune regulatory functions of macrophages and microglia in central nervous system diseases” We can see an acceptable evolution compared to the first version because it has become more structured with more explanation.

the authors have relatively taken the reviewer's remarks and suggestions into consideration, which has positively impacted the quality and consistency of the article.

with this version, the article shows a good scientific level and represents an added value in the research topics that are interested in Immune regulatory functions of macrophages and microglia in central nervous system diseases

the article is accepted for me with this version

Reviewer 3 Report

The authors addressed most of the comments provided. For the suggestion not implemented, they still provided a sufficiently reasonable explanation.